# Computer aided detection for laterally spreading tumors and sessile serrated adenomas during colonoscopy

**Guanyu Zhou**[1☯], **Xun Xiao**[1☯], **Mengtian Tu**[1☯], **Peixi Liu**[1], **Dan Yang**[2], **Xiaogang Liu**[1], **Renyi Zhang**[1], **Liangping Li**[1], **Shan Lei**[1], **Han Wang**[1], **Yan Song**[1], **Pu Wang**[1]*

**1** Department of Gastroenterology, Sichuan Academy of Medical Sciences & Sichuan Provincial People's Hospital, Chengdu, Sichuan, China, **2** Department of Gastroenterology, The Affiliated Hospital of Southwest Medical University, Luzhou, Sichuan, China

☯ These authors contributed equally to this work.
* wangpuhuaxi@qq.com

**Data Availability Statement:** All relevant data are within the manuscript and its Supporting Information files.

## Abstract

### Background

Evidence has shown that deep learning computer aided detection (CADe) system achieved high overall detection accuracy for polyp detection during colonoscopy.

### Aim

The detection performance of CADe system on non-polypoid laterally spreading tumors (LSTs) and sessile serrated adenomas/polyps (SSA/Ps), with higher risk for malignancy transformation and miss rate, has not been exclusively investigated.

### Methods

A previously validated deep learning CADe system for polyp detection was tested exclusively on LSTs and SSA/Ps. 1451 LST images from 184 patients were collected between July 2015 and January 2019, 82 SSA/Ps videos from 26 patients were collected between September 2018 and January 2019. The per-frame sensitivity and per-lesion sensitivity were calculated.

### Results

(1) For LSTs image dataset, the system achieved an overall per-image sensitivity and per-lesion sensitivity of 94.07% (1365/1451) and 98.99% (197/199) respectively. The per-frame sensitivity for LST-G(H), LST-G(M), LST-NG(F), LST-NG(PD) was 93.97% (343/365), 98.72% (692/701), 85.71% (324/378) and 85.71% (6/7) respectively. The per-lesion sensitivity of each subgroup was 100.00% (71/71), 100.00% (64/64), 98.31% (58/59) and 80.00% (4/5). (2) For SSA/Ps video dataset, the system achieved an overall per-frame sensitivity and per-lesion sensitivity of 84.10% (15883/18885) and 100.00% (42/42), respectively.

**Funding:** The author(s) received no specific funding for this work.

**Competing interests:** I have read the journal's policy and the authors of this manuscript have the following competing interests: The CADe system (EndoScreener) was developed by Shanghai Wision AI Co., Ltd. The system was provided free-of-charge for this study. Employees in the company were not involved in the study in any way, including in study design, statistical analysis or manuscript writing. This does not alter our adherence to PLOS ONE policies on sharing data and materials.

## Conclusions

This study demonstrated that a local-feature-prioritized automatic CADe system could detect LSTs and SSA/Ps with high sensitivity. The per-frame sensitivity for non-granular LSTs and small SSA/Ps should be further improved.

## Introduction

Colonoscopy is the gold-standard diagnostic tool for colorectal cancer (CRC) and precancerous lesions. However, the adenoma miss-rate (AMR) ranges from 6% to 27%[1]. It is noteworthy that precancerous lesions with a flat morphology, smooth surface, indistinct boundaries or isochromatic with background are more susceptible to be missed[2, 3, 4]. Given these subtle features of morphology and color, nonpolypoid laterally spreading tumors (LSTs) and sessile serrated adenoma/polyps (SSA/Ps) are among the easist-to-miss lesions[5, 6]. LSTs extend laterally along the colon wall without a polypoid morphology[7, 8]. It could be categorized into 2 types as granular type (LST-G) and non-granular type (LST-NG). The LSTs have a remarkable high risk of malignancy transformation, studies have shown that 63.1% of the investigated LSTs are adenomas with villous structures[6] 20.9%-36.0% of LSTs were found to have high grade intraepithelial neoplasia (HGIN), moreover, LSTs can also develop into a submucosal invasive cancer[6, 9, 10]. Serrated polyps (SPs) are the second most common type of colon polyps, among which, SSA/Ps take about 10–20% of SPs[11, 12] and 5–10%[5] of lesions found during screening colonoscopy. SSA/Ps are a high-risk precursor for CRC via serrated pathway [5, 13, 14], the estimated 10-year risk for CRC transformation from SSA/Ps ranges from 2.56% to 4.43% according to the existence of cytological dysplasia[15] and are dependent on the size [16]. The detection of SSA/Ps via colonoscopy is always difficult due to the pale color and flat morphology[17].

Evidence has shown that with a second party observing the monitor synchronously during colonoscopy, polyp detection rate could be improved by means of addressing the unrecognized polyps within the visual field[1]. However, in comparison with polypoid lesions, to detect LSTs and SSA/Ps is a more challenging task, it is likely that adding an additional human observer would not completely overcome the challenges. Thanks to the breakthrough of artificial intelligence (AI), ideally, real-time computer aided detection (CADe) system during colonoscopy with a comparable detection capability to expert endoscopist could process each frame of the endoscopic video stream and alert any detected targets synchronously, this is a more consistent and reliable set-up than a human assistant[18], because CADe system would consistently analyze every corner of the screen and not miss any single frame of a suspicious lesion. Our CADe system has been previously demonstrated to improve polyp and adenoma detection rate in real world colonoscopy setting[19]. The CADe system was especially developed to capture and identify the local features of colon polyps of variant histology and thus to achieve a high sensitivity and specificity. However, the large-scaled specialized test of such CADe system on LSTs and SSA/Ps is lacking. Therefore, the aim of this study was to investigate the detection performance of the local-feature-prioritized CADe system on LSTs and SSA/Ps.

## Materials and methods

### CADe system (EndoScreener, Shanghai Wision AI Co., Ltd. China)

The real-time automatic polyp detection system[19] (Fig 1) was developed on a deep learning architecture, which was previously validated to have a per-image sensitivity of 94.38%, per-image specificity of 95.92% and an area under the receiver operating characteristic curve of

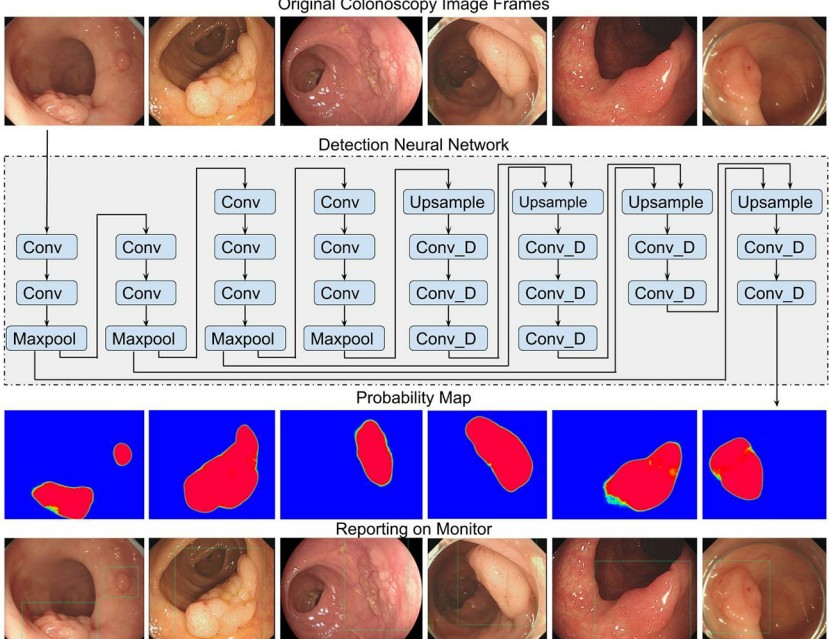

**Fig 1. Schematic of the automatic polyp detection algorithm [19].** a. Original colonoscopy image frames generated during regular colonoscopy procedures. b. Deep convolutional neural network (CNN): SegNet architecture. (http://mi. eng.cam.ac.uk/projects/segnet/), which calculates the probability of. belonging to a polyp for each pixel in the input colonoscopy image frame. c. Probability map, which showed the probability of belonging to a polyp (color blue represents probability = 0, color red represents probability = 1, color in between represents 0<probability<1), for each pixel in the input image frame. d. Based on the probability map, hollow blue boxes are added to the original image, in order to highlight the polyp areas for the observing clinicians.

0.984 on consecutive colonoscopy images. In addition, the algorithm has a per-polyp sensitivity of 100.00% (per-image sensitivity of 91.64%) and a per-image specificity of 95.40% in colonoscopy videos.

In the preliminary study, the algorithm was trained on 5545 colonoscopy images with variant polyp morphology and histology, among which there were 103 images of 69 LSTs, including 56 images of 34 LST-G and 47 images of 35 LST-NG.

To better distinguish the intra-class and inter-class variability and thus to achieve high sensitivity, specificity and to effectively detect partially occluded polyps, this network structure of the algorithm is characterized by its small receptive fields that extract the local visual features of the input image, as well as the high nonlinearity of its consecutive convolution layers without pooling[19].

The system provides a per-pixel labeling of its detected area on each processed frame.

## Detection validation

This study was conducted in the endoscopy center of Sichuan provincial people's hospital, China.

We retrospectively collected images of LSTs and video clips of SSA/Ps from the database of our center which were not included for the development of EndoScreener during 2017 from 2 endoscopy centers. 1451 images of 199 LSTs from 184 patients were obtained from July 2015 to January 2019. 82 videos of SSA/Ps based on pathological diagnosis from 26 patients were obtained from September 2018 to January 2019, each SSA/Ps video was cut into clip from the

first frame of its appearance till the last frame it disappeared. The average length of the videos was 17.99-second-long. All the images and videos in the database were acquired from Olympus EVIS LUCERA CV260 (SL)/CV290 (SL) and Fujifilm VP-4450 HD and adapted high-quality and high definition colonoscopes.

Basic demographic characteristics including gender and age; the sizes, locations and histological results of each lesion were illustrated.

We used the CADe system to process each image of LSTs dataset and video clips of SSA/Ps datasets, the system provided a per-pixel labeling of its detected area. The evaluation of the algorithm is based on the consensus among a 3-person panel of senior endoscopists in our center who carefully re-checked each image of the labeled LSTs dataset and each frame of the labeled SSA/Ps video dataset. For image analysis on LSTs, the panel of experienced endoscopists recorded the number of correctly labeled LSTs missed LSTs. Then the panel labeled the border of each LST by hand. For SSA/Ps video assessment, the panel of experienced endoscopists carefully re-checked each labelled frame in the video and recorded the number of correctly labeled polyps and missed polyps. Authors had no access to information that could identify individual participants during or after data collection.

This study was reviewed and approved by institutional review board (Ethics committee) of Sichuan Academy of Medical Sciences & Sichuan Provincial People's Hospital on May 17, 2017.

Written informed consents were obtained from all individuals who were detected LSTs and SSA/Ps during colonoscopy and approved to provide the lesions' images and videos for this study.

All methods were carried out in accordance with relevant regulations of Sichuan Academy of Medical Sciences & Sichuan Provincial People's Hospital.

The study protocol conforms to the ethical guidelines of the 1975 Declaration of Helsinki as reflected in a priori approval by the institution's human research committee.

## Statistical analysis

The main outcome is per-image or per-frame sensitivity and per-lesion sensitivity. The secondary outcome is ratio of correctly labeled area.

If the label of algorithm detection is on an actual lesion (LSTs or SSA/Ps according to the tested datasets), then it is considered a true positive (TP) and only one true positive will be counted for each actual lesion, regardless of how many algorithm labels fall on that lesion. The absence of the algorithm label on an actual lesion is counted as one false negative (FN). Therefore, the total number of true positives and false negatives is the total number of lesion appearance in the validation images or frames. Therefore, per-frame sensitivity was defined as TP divided by total number of polyp appearances = TP / (TP+FN). This is the commonly accepted statistical method for evaluating image detection[20].

Because in real clinical setting, intermittent detections could also lead to an effective notice to the endoscopist and thus to avoid a miss diagnosis, we used per-lesion sensitivity to show the pragmatic detection capability of the CADe system. The per-lesion sensitivity is defined as the number of lesions correctly detected by the algorithm in at least one frame divided by the total number of actual lesions.

To measure how well the algorithm can detect the full range of an LST, we also measured the coverage ratio of correctly labeled area, which is defined as the number of pixels of the correctly labeled area which were confirmed by the endoscopist-panel, divided by the number of pixels of the panel-labeled area of each lesion on each frame.

## Results

### Baseline data of the tested datasets (Table 1)

In the LSTs image dataset, there were 1451 images containing LSTs of 199 LSTs lesions from 184 patients, among which 90 (50.00%) were female, and the mean age was 63.11±11.47(standard deviation, s.d.). There were 24 carcinomas (12.06%), 16 SSA/Ps (8.04%), 148 conventional adenomas (74.37%), 77 advanced adenomas (38.69%), and 11 hyperplastic or inflammatory component (5.53%). The average diameter of all the LSTs was 2.33±1.13cm.

In the SSA/Ps Video Dataset, there were 82 colonoscopy video clips of 42 SSA/Ps lesions with 18885 frames in total, each with an SSA/Ps appearing from the beginning until the end. The length of the dataset is 12.59 minutes in total and 17.99s per lesion on average. These videos were obtained from 26 patients, including 7 females (26.92%). The mean age was 50.81±10.07. All of the SSA/Ps were Is in morphology according to Paris classification system.

In the SSA/Ps video dataset, there are 29 (69.05%) diminutive lesions ($\leq$0.5cm), 12 (28.57%) small lesions ($<$0.5cm, $<$ = 1cm), and one (2.38%) large lesion. Other baseline information are presented in Table 1.

### The detection sensitivity and ratio of correctly labeled area on LSTs (Table 2)

LSTs were categorized into 4 subgroups which are LST-G(H), LST-G(M), LST-NG(F), and LST-NG(PD), the per-image sensitivity of each subgroup was 93.97% (343/365), 98.72% (692/701), 85.71% (324/378) and 85.71% (6/7), while the per-lesion sensitivity of each subgroup was 100.00% (71/71), 100.00% (64/64), 98.31% (58/59) and 80.00% (4/5). The overall per-image sensitivity and per-lesion sensitivity of LSTs were 94.07% (1365/1451) and 98.99% (197/199) respectively. 2 lesions were missed by CADe system in LST-NG(F) and LST-NG(PD) subgroup separately.

**Table 1. Baseline information.**

| | LST Images Dataset | SSA Videos Dataset |
|---|---|---|
| **Data acquisition**[a] | July2015-January2019 | September2018- January 2019 |
| **Content**[b] | 1451 images containing LST | 82 colonoscopy video clips, each with a SSA appearing from the beginning until the end. 12.59 min in total and 17.99s per polyp on average. |
| **Device**[c] | Olympus and Fujifilm | Olympus and Fujifilm |
| **Patient demographics** | 184 patients,92(50.00%) female; age, mean(s.d.):63.11(11.47) | 26 patients,7(26.92%) female; age, mean(s.d.):50.81(10.07) |
| **Polyp histology** | Total LST number 199(100%) Carcinoma 24(12.06%) SSAP 16(8.04%) Adenomatous 148(74.37%) Advanced Adenoma 77(38.69%) Hyperplastic and Inflammatory 11(5.53%) | Total SSA/Ps number 42(100%) SSAP 42(100%) |
| **Polyp location** | Rectum 76(38.19%) Sigmoid colon 25(12.56%) Descending colon, including splenic flexure 11(5.53%) Transverse colon 31(15.58%) Ascending colon, including hepatic flexure 42(21.11%) Cecum 14(7.04%) | Rectum 9(21.43%) Sigmoid colon 11(26.19%) Descending colon, including splenic flexure 3(7.14%) Transverse colon 11(26.19%) Ascending colon, including hepatic flexure 7(16.67%) Cecum 1 (2.38%) |
| Polyp size (cm) | size, mean(s.d.):2.33(1.13) | Small ($\leq$0.5) 29(69.05%) Moderate ($>$0.5&$<$ = 1) 12(28.57%) Large ($>$1) 1(2.38%) |

[a]All datasets were acquired from the Endoscopy Center of Sichuan Provincial People's Hospital of China and The Affiliated Hospital of Southwest Medical University.

[b]Resolution of images and videos are 704 × 576, 1,920 × 1,080 or 1,280 × 1,024.

[c]Olympus EVIS LUCERA CV260 (SL)/CV290 (SL) and Fujifilm 4450 HD. NA.

**Table 2. Detection sensitivity for LSTs.**

| LST Type | LST-G(H) | LST-G(M) | LST-NG(F) | LST-NG(PD) | Total |
|---|---|---|---|---|---|
| **Per-image sensitivity** | 93.97% | 98.72% | 85.71% | 85.71% | 94.07% |
| **Per-lesion sensitivity** | 100.00% | 100.00% | 98.31% | 80.00% | 98.99% |
| **Labeled Area** | 35.66% | 41.49% | 35.6% | 43.47% | 39.41% |

The ratio of correctly labeled area for each subgroup was LST-G(H) 35.66%, LST-G(M) 41.49%, LST-NG(F) 35.6%, LST-NG(PD) 43.47%.

### The detection sensitivity on SSA/Ps (Table 3)

According to the size, the SSA/Ps was classified into 3 subtypes: diminutive ($\leq$0.5cm), small (>0.5cm&$\leq$1cm) and large (>1cm), the CADe system had achieved a per-frame sensitivity of each subgroup as 80.29% (10356/12899), 92.90% (5080/5468), 86.29% (447/518). The per-lesion sensitivity for each subgroup was 100.00% (29/29), 100.00% (12/12) and 100.00% (1/1) respectively. The overall per-frame sensitivity and per-lesion sensitivity of SSA/Ps were 84.10% (15883/18885) and 100.00% (42/42).

## Discussion

CRC is one of the leading causes of cancer related death[21, 22] and is a major public health issue given its high incidence and mortality rate. Screening colonoscopies have allowed a significant decrease of mortality rate and incidence of CRC in adults (by 51% and 32%, respectively). colonoscopy has also allowed an increase in 5-year survival rate in CRC, mainly as a result of early detection as well as removal of precancerous lesions[23]. Missed precancerous lesions, including conventional adenomas, SSA/Ps as well as LSTs with various pathology, during colonoscopy might lead to subsequent colorectal cancer. Miss diagnosis of any precancerous lesions should be decreased because evidence has shown that less interval cancer could be expected under endoscopists with high adenoma detection rate (ADR)[24, 25]. Although conventional adenomatous polyps with a protruding morphology and redder color contributes the majority of ADR, the detection of non-polypoid LSTs and isochromatic SSA/Ps is equally important because they are even more susceptible to be missed[6]and some of them are more risky for malignant transformation. Given the importance of better identification of any precancerous lesions during colonoscopy, tremendous efforts from both hardware engineers and clinicians have been put to make lesions more visible to the endoscopists[26], However, endoscopists still miss a large portion of precancerous lesions within the screen, for both lesions with non-obvious visual features and lesions that briefly flashed across the screen. These conditions are very challenging even for experienced endoscopists, because humans are always susceptible to "inattentional blindness", "change blindness", visual gaze pattern, fatigue, inter-observer variabilities or any distractions [18, 27, 28, 29, 30, 31, 32]. Till now, only second-observer strategies seem helpful to decrease the miss rate of visible polyps for low ADR detectors[26, 33]. However, adding a second human observer might not completely overcome the challenges.

**Table 3. Detection sensitivity for SSA/Ps.**

| SSAP Size | Small ($\leq$0.5) | Moderate (>0.5&<=1) | Large (>1) | Total |
|---|---|---|---|---|
| **Per-frame sensitivity** | 80.29% | 92.90% | 86.29% | 84.10% |
| **Per-lesion sensitivity** | 100.00% | 100.00% | 100.00% | 100.00% |

In comparison with human observer, high-performance CADe has high reproducibility, fidelity and uniformity[34]. These advantages are especially important for detecting lesions that are non-obvious, briefly visible or partially occluded.

Therefore, it is crucially important to investigate the detection performance exclusively on LSTs and SSA/Ps of any CADe system which was developed for screening early cancers and pre-cancerous lesions, because if a CADe system is only capable to detect polypoid lesions, then its clinical value for decreasing interval cancer is inadequate[19]. In the preliminary technical study, we have considered the importance of identifying the local features of a lesion, as pattern of pits and micro-vessels, and thus to build an algorithm that relies less on overall morphology[19] This strategy is especially helpful for identifying flat lesions. Moreover, when a lesion flashed across the edge of the screen or partially behind a fold, an algorithm that is trained more on local features and does not rely on the full appearance may still detect it by capturing the detailed features[19]. Therefore, such CADe system should be assumed to have a good performance in detecting LSTs and SSA/Ps.

In this study, it is easy for the CADe system to detect the LST-G due to multiple polypoid granules. As a result, the 93.97% per-frame sensitivity for LST-G-H (homogeneous type) and 98.72% per-frame sensitivity for LST-G-M (nodular mixed type) was comparable with the per-frame sensitivity for all kinds of polyps in our preliminary study[19]. However, LST-NG is difficult to detect because of the absence of any polypoid protuberance. Nevertheless, the surface texture of LST-NG is distinguishable because most of the LSTs are conventional adenomas in pathology[6] and should have the same micro-surface features. The CADe system achieved an 85.71% per-frame sensitivity for LST-NG-F (flat type), 85.71% for LST-NG-PD (pseudo-depressed type), the system missed one lesion in each subgroup.

The ratio of correctly labeled area for the 4 subgroups ranges from 35.6% to 43.47%. The CADe system detected well on the protruding part and the margin. Moreover, when lesion is observed at a close distance, detailed micro-surface structures are identifiable, the CADe system could also detect well on the smooth regions (Figs 2 and 3). However, the CADe system failed to completely label the smooth area when a lesion was photographed in far distance when micro-surface structures is visually indistinguishable, it was also challenging for the system to detect the smooth area when the color of the lesion was similar to the background. This is attributed to inadequate visual features. New technology and detection strategy are needed to be deployed to better identify flat and isochromatic lesions that lie far from the camera.

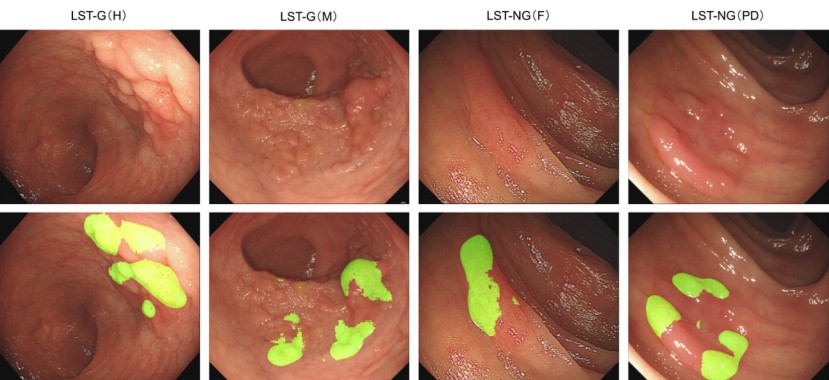

**Fig 2. Detection labeling of CADe system on LSTs with different morphology.** Green tags are per-pixel predictions of the system in 4 subgroups of LSTs.

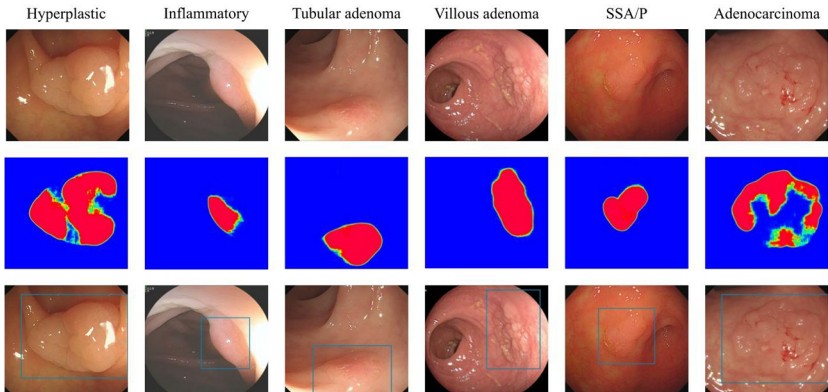

**Fig 3. Detection labeling of CADe system on LSTs with different pathology.** Green tags are per-pixel predictions of the system on LSTs with different pathology, as hyperplastic, inflammatory, tubular adenoma, villous adenoma, serrated adenoma and adenocarcinoma.

Noteworthy, although partially tagged, the alerts generated by the CADe system may still effectively notice the endoscopist and avoid a miss during real clinical setting.

SSA/Ps are usually flat with an unclear boundary, the color the SSA/Ps are commonly whitish or isochromatic to the background, the micro-surface structures are similar with hyperplastic polyps under white light colonoscopy[35, 36]. Hence, the overall per-frame sensitivity was no more than 90%, the 80.29% sensitivity (S1–S4 Videos) for small SSA/Ps subgroup was much lower than the 91.64% sensitivity for all kinds of polyps in the preliminary study[19]. The per-lesion sensitivity was 100.0%, nevertheless, the system might miss a briefly visible lesion because it misses one fifth of the frames. Therefore, it is important to improve the detection performance by adding more training data for the CADe system to "learn" the unique II-O pit and varicose microvascular vessel (VMV)[37] of SSA/Ps. Nevertheless, the performance was consistent with the previous technical study, the same CADe system with such a sensitivity for SSA/Ps has increased the deetction rate of diminutive polyps significantly[4][18].

However, although adding more training data might be helpful for the CADe system to distinguish subtle inter-class visual features, it will be extremely difficult to obtain a marginal benefit as well as to suppress false positive rate, because when a lesion lies far from the camera, it is challenging for any CADe system or human expert to be specific due to inadequate pixels in the image. Further improvement may be obtained by double-AI CADe system, of which one AI captures local feature of definite lesions which lie close, a second AI independently captures other features of suspected lesions which lie far. Therefore, such double-AI strategy should be more specific for definite lesions and won't compromise the sensitivity for suspected lesion which lies far. Moreover, the distinguished display of definite and suspected lesions could be more comprehensible, and endoscopists can rely on the new system instead of being confused by false alarms or dissatisfied by the insufficient sensitivity for suspected lesions.

In addition, it is important to notice that the current CADe system should be used as a supplement quality assurance tool, because none of them has achieved 100% accuracy. The endoscopists should carefully inspect the colon firstly by themselves instead of fully relying on the detection of the system. The users should be aware of the system-undetected lesions with a basic attention. Therefore, it is crucially important for a CADe system to be well developed and rigorously validated. On one hand, if and only if a CADe system has a good performance to detect non-obvious lesions, it could be considered clinical helpful. On the other hand, the validation dataset of such system should be representative with a large sample size and various target lesions. Moreover, the validation dataset should be obtained from real clinical setting

and should contains consecutive unselected unaltered images or videos after the development of the algorithm. Only this, the result, i.e sensitivity and specificity, of the system's performance is meaningful[19] and the endoscopist will accurately know how much he can rely on a CADe system. Last but not the least, a post-colonoscopy double check of the system labeled videos by senior endoscopists should be considered to make sure full benefit of the patients.

This study has several limitations, firstly, there were not sufficient samples of LST-NG-PD images and large size SSA/Ps videos, the detection result might not be generalizable for these subtypes, more data should be added to evaluating the detection performance of the CADe system on these subtypes. Secondly, as a limitation of current calculation power of GPU, the current system can only extract a small portion of pixels from these lesions and thus might miss some detailed characteristics after extraction. Therefore, the system might not be so sensitive for lesions lie far from the camera when the number of pixels is inadequate, this will be addressed along with the hardware development. Thirdly, we did not assess the specificity of the system in this study, because the algorithm was unchanged as in the preliminary study [19], in which the specificity has been tested on larger image and video datasets.

## Conclusion

The local-feature-prioritized polyp detection CADe system could detect LSTs and SSA/Ps with a high sensitivity which indicates it might be a promising quality assurance during colonoscopy to reducing miss diagnosis of easy-to-miss precancerous lesions along with conventional polyps and thus has the potential to decrease the risk of interval cancer. The impact of such CADe system on the detection of LSTs and SSA/Ps in real clinical setting should be specifically investigated in large-scaled clinical trials.

## Supporting information

**S1 Video.**
(AVI)

**S2 Video.**
(AVI)

**S3 Video.**
(AVI)

**S4 Video.**
(AVI)

## Acknowledgments

Dr. Xiao Xiao, Zhiwei Zhang and Jingjia Liu from Shanghai Wision AI Co., Ltd. developed the CADe system according to the study design and provided technical description of the technology. We thank Dr. Wenfei Zhang for the advice on statistical analysis.

## Author Contributions

**Conceptualization:** Pu Wang.

**Data curation:** Mengtian Tu, Peixi Liu, Dan Yang, Xiaogang Liu, Renyi Zhang, Liangping Li, Shan Lei, Han Wang, Yan Song.

**Methodology:** Guanyu Zhou, Pu Wang.

**Writing – original draft:** Guanyu Zhou, Xun Xiao, Mengtian Tu, Pu Wang.

**Writing – review & editing:** Guanyu Zhou, Xun Xiao, Pu Wang.

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
