## [Decision Letter · Decision Letter 0]

12 Mar 2020

PONE-D-19-36040

Computer aided detection for laterally spreading tumors and sessile serrated adenomas during colonoscopy

PLOS ONE

Dear Pu Wang,

Thank you for submitting your manuscript to PLOS ONE. After careful consideration, we feel that it has merit but does not fully meet PLOS ONE’s publication criteria as it currently stands. Therefore, we invite you to submit a revised version of the manuscript that addresses the points raised during the review process.

We would appreciate receiving your revised manuscript by Apr 26 2020 11:59PM. To enhance the reproducibility of your results, we recommend that if applicable you deposit your laboratory protocols in protocols.io, where a protocol can be assigned its own identifier (DOI) such that it can be cited independently in the future. For instructions see: http://journals.plos.org/plosone/s/submission-guidelines#loc-laboratory-protocols

We look forward to receiving your revised manuscript.

Kind regards,

Wajid Mumtaz

Academic Editor

PLOS ONE

Journal Requirements:

3. In the ethics statement in the Methods and online submission information, please ensure that you have specified the form of consent that you obtained (for instance, written or verbal, and if verbal, how it was documented and witnessed).

4. Thank you for stating the following in the Competing Interests section: "I have read the journal's policy and the authors of this manuscript have the following competing interests:The CADe system (EndoScreener) was developed by Shanghai Wision AI Co., Ltd. The system was provided free-of-charge for this study. Employees in the company were not involved in the study in any way, including in study design, statistical analysis or manuscript writing."

6. Please amend either the title on the online submission form (via Edit Submission) or the title in the manuscript so that they are identical.

Reviewers' comments:

Reviewer's Responses to Questions

**Comments to the Author**

1. Is the manuscript technically sound, and do the data support the conclusions?

Reviewer #1: Yes

2. Has the statistical analysis been performed appropriately and rigorously? 

Reviewer #1: I Don't Know

3. Have the authors made all data underlying the findings in their manuscript fully available?

Reviewer #1: Yes

4. Is the manuscript presented in an intelligible fashion and written in standard English?

Reviewer #1: Yes

5. Review Comments to the Author

Reviewer #1: Colonoscopy is the gold standard for detecting and removing lesions to prevent cancer. Unfortunately, the miss rate can be as high as 25% even for adenomatous polyps and even higher for LST and SSA/P. The authors have developed an automated system to detect lesions. They test whether this system permits LST and SSA/P to be reliably detected. The data indicate the system improves detection but still needs to be perfected. A few concerns need to be addressed.

Major

1) For a broad audience, much more information needs to be provided regarding the basics of colorectal cancer and image analysis. For example, a figure showing different pathologies would be invaluable.

2) Define all terms and abbreviations when first used.

3) The Discussion should address how this work could impact clinical practice.

Minor

1) Intro - "alone" should be "along"

2) Intro - "literatures" should be "studies"

3) Methods - averagely is not a word

4) Discussion - "caner" should be "cancer"

6. PLOS authors have the option to publish the peer review history of their article (what does this mean?). If published, this will include your full peer review and any attached files.

Reviewer #1: No

---

## [Author Response · Author response to Decision Letter 0]

23 Mar 2020

• A rebuttal letter that responds to each point raised by the academic editor and reviewer(s). This letter should be uploaded as separate file and labeled 'Response to Reviewers'.

• A marked-up copy of your manuscript that highlights changes made to the original version. This file should be uploaded as separate file and labeled 'Revised Manuscript with Track Changes'.

• An unmarked version of your revised paper without tracked changes. This file should be uploaded as separate file and labeled 'Manuscript'.

Answer: we have modified the manuscript according to PLOS ONE style.

Answer: Thanks for the recommendation. We’ve made some edits to the manuscript, hopefully it is now with decent English. If language is the last problem there, we can have it further finetuned. 

2. In the ethics statement in the Methods and online submission information, please ensure that you have specified the form of consent that you obtained (for instance, written or verbal, and if verbal, how it was documented and witnessed).

Written informed consents were obtained from all participants, as modified in the revised manuscript

3. Thank you for stating the following in the Competing Interests section: "I have read the journal's policy and the authors of this manuscript have the following competing interests:The CADe system (EndoScreener) was developed by Shanghai Wision AI Co., Ltd. The system was provided free-of-charge for this study. Employees in the company were not involved in the study in any way, including in study design, statistical analysis or manuscript writing."

We have added this section according to your requirement 

We have added the statements in the manuscript according to your requirement

We have added a data availability section to declare where the minimal and entire data set underlying the results described in the manuscript can be found

6. Please amend either the title on the online submission form (via Edit Submission) or the title in the manuscript so that they are identical.

We have confirmed the title as “Computer aided detection for laterally spreading tumors and sessile serrated adenomas during colonoscopy”

Reviewers' comments:

Reviewer's Responses to Questions

Comments to the Author

1. Is the manuscript technically sound, and do the data support the conclusions?

Reviewer #1: Yes

2. Has the statistical analysis been performed appropriately and rigorously? 

Reviewer #1: I Don't Know

 Answer: the statistical analysis of the study is a basic validation of sensitivity and specificity, these are very routine metrics, the definitions are consistent with our previous study which published at “Nature Biomedical Engineering 2018;2:741–748”, the methodology and definitions are also consistent with other major studies around AI-medicine filed. etc. 

Wang P, Xiao X, Glissen Brown JR, et al. Development and validation of a deep-learning algorithm for the detection of polyps during colonoscopy. Nature Biomedical Engineering 2018;2:741–748

Bernal, J. et al. Comparative validation of polyp detection methods in video colonoscopy: results from the MICCAI 2015 endoscopic vision challenge. IEEE Trans. Med. Imaging 36, 1231–1249 (2017).

Bandos AI, Rockette HE, Song T, Gur D. Area under the free response ROC curve (FROC) and a related summary index. Biometrics 2009;65:247–256

3. Have the authors made all data underlying the findings in their manuscript fully available?

Reviewer #1: Yes

4. Is the manuscript presented in an intelligible fashion and written in standard English?

Reviewer #1: Yes

5. Review Comments to the Author

Reviewer #1: Colonoscopy is the gold standard for detecting and removing lesions to prevent cancer. Unfortunately, the miss rate can be as high as 25% even for adenomatous polyps and even higher for LST and SSA/P. The authors have developed an automated system to detect lesions. They test whether this system permits LST and SSA/P to be reliably detected. The data indicate the system improves detection but still needs to be perfected. A few concerns need to be addressed.

Major

1) For a broad audience, much more information needs to be provided regarding the basics of colorectal cancer and image analysis. For example, a figure showing different pathologies would be invaluable.

Answer: thanks for the comments. For a broad audience, we added a background introduction at the beginning of the discussion section, as “CRC is one of the leading causes of cancer related death[ ] [ ] and is a major public health issue given its high incidence and mortality rate. Screening colonoscopies have allowed a significant decrease of mortality rate and incidence of CRC in adults (by 51% and 32%, respectively). colonoscopy has also allowed an increase in 5-year survival rate in CRC, mainly as a result of early detection as well as removal of precancerous lesion”

and a figure 3 which illustrates the detection performance of CADe system on LSTs with different pathologies.

Fig 3

2) Define all terms and abbreviations when first used.

Answer: we confirm all terms and abbreviations were defined 

3) The Discussion should address how this work could impact clinical practice.

Answer: we added “Noteworthy, although partially tagged, the alerts generated by the CADe system may still effectively notice the endoscopist and avoid a miss during real clinical setting. ” to show the impact of the CADe system on LSTs detection during colonoscopy; we added “Nevertheless, the performance was consistent with the previous technical study, the same CADe system with such a sensitivity for SSA/Ps has increased the deetction rate of diminutive polyps significantly” to show the impact of the CADe system on SSA/Ps detection during colonoscopy

Minor

1) Intro - "alone" should be "along" 

2) Intro - "literatures" should be "studies" 

3) Methods - averagely is not a word 

4) Discussion - "caner" should be "cancer" 

All the typos have been corrected

6. PLOS authors have the option to publish the peer review history of their article (what does this mean?). If published, this will include your full peer review and any attached files.

Do you want your identity to be public for this peer review? For information about this choice, including consent withdrawal, please see our Privacy Policy.

Reviewer #1: No

[1] American Cancer Society. Cancer Facts and Figures: 2017. Atlanta, Georgia. 2017. https://www.cancer.org/research/cancer-facts-statistics/all-cancer-facts-figures/cancer-facts-fgures-2017.html. Accessed August 15, 2017

[2] Fang JY, Zheng S, Jiang B, et al. Consensus on the Prevention, Screening, Early Diagnosis and Treatment of Colorectal Tumors in China: Chinese Society of Gastroenterology, October 14-15, 2011, Shanghai, China. Gastrointest Tumors 2014;1:53-75

---

## [Decision Letter · Decision Letter 1]

3 Apr 2020

Computer aided detection for laterally spreading tumors and sessile serrated adenomas during colonoscopy

PONE-D-19-36040R1

Dear Dr. Wang,

We are pleased to inform you that your manuscript has been judged scientifically suitable for publication and will be formally accepted for publication once it complies with all outstanding technical requirements.

With kind regards,

Wajid Mumtaz

Academic Editor

PLOS ONE

Additional Editor Comments (optional):

Reviewers' comments:

Reviewer's Responses to Questions

**Comments to the Author**

1. If the authors have adequately addressed your comments raised in a previous round of review and you feel that this manuscript is now acceptable for publication, you may indicate that here to bypass the “Comments to the Author” section, enter your conflict of interest statement in the “Confidential to Editor” section, and submit your "Accept" recommendation.

Reviewer #1: All comments have been addressed

2. Is the manuscript technically sound, and do the data support the conclusions?

Reviewer #1: Yes

3. Has the statistical analysis been performed appropriately and rigorously? 

Reviewer #1: Yes

4. Have the authors made all data underlying the findings in their manuscript fully available?

Reviewer #1: Yes

5. Is the manuscript presented in an intelligible fashion and written in standard English?

Reviewer #1: Yes

6. Review Comments to the Author

Reviewer #1: (No Response)

7. PLOS authors have the option to publish the peer review history of their article (what does this mean?). If published, this will include your full peer review and any attached files.

Reviewer #1: No

---

## [Editor Report · Acceptance letter]

8 Apr 2020

PONE-D-19-36040R1 

Computer aided detection for laterally spreading tumors and sessile serrated adenomas during colonoscopy 

Dear Dr. Wang:

I am pleased to inform you that your manuscript has been deemed suitable for publication in PLOS ONE. Congratulations! Your manuscript is now with our production department. 

With kind regards,

on behalf of

Dr. Wajid Mumtaz 

Academic Editor

PLOS ONE